# OpenReview forum: "Hybrid Preference Optimization: Augmenting Direct Preference Optimization with Auxiliary Objectives"
_ICLR.cc/2025/Conference — ICLR 2025 Conference Withdrawn Submission_

### Official Review · Reviewer_EMmv · 2024-11-01

**Soundness:** 3
**Presentation:** 3
**Contribution:** 2
**Rating:** 5
**Confidence:** 4

**Summary:**

This work proposed Hybrid Preference optimization which optimizes the human preference along side with weighted auxiliary rewards, e.g., toxicity, readability, etc. Specifically, the authors augment the preference loss with an advantage-weighted maximum likelihood objective and use expectile regression to train the value network. In the experiment, the authors consider several auxiliary objectives, e.g., reading level and safety.

**Strengths:**

The author conduct extensive experiments in the setting of preference learning with auxiliary objectives, together with several ablation studies on effect of varying hyperparameters, reward weights.

**Weaknesses:**

1. The proposed method introduces an additional term in the objective to optimize auxiliary rewards, while most of the baselines only optimize towards the preference dataset. There could be straight-forward approaches to incorporate the auxiliary reward to the single-objective baselines, e.g., fit a reward model on the compound reward and use it to construct preference pairs. Further, the authors also barely discuss their choice of the auxiliary loss with other variants (see point 2).

2. Optimizing the reverse KL in equation (8) in offline setting is investigated in [1], where using self-normalized importance sampling with proper weight leads to better performance than optimizing the forward KL. The authors should discuss and compare with this related approach.

3. A crucial aspect of multi-objective alignment is to evaluate the frontier of multiple objectives. However, the paper did not compare with the multi-objective baselines in terms of this aspect.

[1] Ji, Haozhe, et al. "Towards efficient and exact optimization of language model alignment." ICML (2024).

**Questions:**

1. Could the authors incorporate the auxiliary rewards into the preference learning baselines for a fair comparison?

2. Could the authors compare with other variants of implementing the auxiliary objective, e.g., [1] that directly optimizes the reverse KL.

3. Could the authors compare their method with multi-objective baselines in terms of trade-offs among objectives?

---

> ### Author Response · Authors · 2024-11-14
> **Rebuttal**
>
> Thank you for your feedback. Below are our responses to your comments.
>
> > The proposed method introduces an additional term in the objective to optimize auxiliary rewards, while most of the baselines only optimize towards the preference dataset.
>
> There are 3 baselines (aoPPO, MODPO, and A-LOL) that optimize for the same auxiliary objectives with the same weight as HPO. There are 4 non-preference baselines for two reasons: they are significantly more recognizable than multi-objective methods and help ground the reality that our method equals or outperforms the most popular alignment methods, and they also serve as a point of comparison to their multiobjective counterparts (e.g., as in line 454, where we compare DPO to MODPO). In most cases, aoPPO does not optimize auxiliary rewards better than PPO (Table 2), nor does MODPO compared to DPO. The existing MO techniques simply do not optimize the objectives well enough, overall.
>
> While there are other multi-objective approaches such as DRO, they are quite similar to multi-objective approaches we have already compared to (e.g., A-LOL's policy gradient is nearly identical to that of DRO). If you have any suggestions on multi-objective approaches that are distinct enough from the ones we have tested already, please let us know.
>
> >There could be straight-forward approaches to incorporate the auxiliary reward to the single-objective baselines, e.g., fit a reward model on the compound reward and use it to construct preference pairs.
>
> That is possible, but this is a theoretically weak setup for optimizing for a set of multiple objectives with different weights (i.e., that essentially prescribe a choice from an enumerated set of ranked outcomes), as we show in Section 4.1. Binary outcomes are simply insufficient to capture the complexity of balancing multiple rewards.
>
> > Optimizing the reverse KL in equation (8) in offline setting is investigated in [1], where using self-normalized importance sampling with proper weight leads to better performance than optimizing the forward KL.
>
> We explain this briefly on line 838 (Appendix B) and in the related work. It doesn't meet our criteria for several reasons:
> - If we want to optimize the reverse KL directly, we must sample from the LLM since the expectation will not be over the data distribution. This is intractable and would result in training times upward of 2 weeks for large LLMs, even with many tricks (line 340).
> - Otherwise, generally, importance sampling is unstable (per Baheti et al., 2023 and as stated on line 99) and there are many RL instances in which it performs poorly. One example is A-LOL, where we use importance sampling/weights, and there are tricks such as clipping required. Even still, that baseline does not perform well and is one of the least safe alignment approaches.
> - Finally - importance weights that leverage the reference model are still biased because $\pi_\beta \ne \pi_{\hat{\beta}}$ (i.e., the behaviourally cloned LLM is not unbiased relative to the true data distribution). Given that the (log) probability of generating the true outputs in the data distribution by $\pi_{\hat{\beta}}$ are low, which is why perplexity is never that close to 1, it indicates that there is likely a non-negligible and significant amount of bias.
>
> Our fundamental goal is to avoid sampling from the LLMs (lines 66 & 106), importance sampling and too many tricks (like clipping, multiple Q/V-networks, explicit conservative penalties), as stated on line 254.
>
> >Could the authors incorporate the auxiliary rewards into the preference learning baselines for a fair comparison?
>
> MODPO is a version of DPO with the auxiliary rewards, and aoPPO is a version of oPPO with the auxiliary rewards. These two multi-objective baselines certainly are fair comparisons. A-LOL is a technique for preference learning proposed in Baheti et al. (2023).
>
> >Could the authors compare their method with multi-objective baselines in terms of trade-offs among objectives?
>
> We consider tradeoffs with objectives in Table 4, and these numbers completely dominate any achieved by MODPO or A-LOL across any possible weight selection. There is a clear increase in readability as the weight for it is increased, and it seems as if the toxicity increases consequently (which is expected).
>
> We are currently running experiments with different weights for aoPPO (the only competitive multi-objective approach in our experiments) and will construct a Pareto front. Do you think that would improve your assessment of our work?

---

> > ### Comment · Reviewer_EMmv · 2024-11-25
> > **Official Comment by Reviewer EMmv**
> >
> > Thanks for the author's clarification, my only concern is the comparison with the multi-objective baselines. A figure showing that HPO dominates other multi-objective baselines in terms of the frontier on two contradicting metrics, e.g., toxicity and readability, is missing. I would like to raise my score if the author present the result.

---

### Official Review · Reviewer_aNak · 2024-11-03

**Soundness:** 2
**Presentation:** 2
**Contribution:** 2
**Rating:** 3
**Confidence:** 4

**Summary:**

In this paper, the authors propose a new multi-objective preference optimization method. The main advantage of this method is that it is a one-step fine-tuning method that performs well on multiobjectives. They compare this method with offline reinforcement learning methods like oPPO and direct preference methods like KTO and DPO. They indicate that this method outperforms others on all objectives.

**Strengths:**

It is interesting to propose a new multiobjective direct preference optimization method. This paper also focuses on broad experiments and analysis, which is the main strength of this paper.

**Weaknesses:**

Although the authors performed impressively on different benchmarks, I have some concerns. I would be happy to discuss them with the authors further.

1. **Lack of comparision**. The first concern is about the methods selected for comparison. I think the authors need to select better methods that are aligned with their hypothesis, like safe-RLHF. Also, The proposed method is similar to the Direct Reward Optimization (DRO) method. It would be great if the authors considered these methods as competitors.

2. **Old models**. Another concern is outdated models. I suggest using the new versions of the LLaMA, Mistral, or Gemma-2 models.

3. **Lack of exploration on hyperparameters**.DPO, KTO, and other optimization methods are very sensitive to different hyperparameters like beta, batch size, and learning rate. So, I encourage the authors to compare the methods using their best hyperparameters.

---
safe-RLHF: https://arxiv.org/abs/2310.12773

DRO: https://arxiv.org/abs/2405.19107

**Questions:**

All concerns and suggestions are mentioned in the weakness section.

---

> ### Author Response · Authors · 2024-11-12
> **Rebuttal**
>
> Thank you for your feedback. Below are our responses to your comments.
>
> >I think the authors need to select better methods that are aligned with their hypothesis, like safe-RLHF. Also, The proposed method is similar to the Direct Reward Optimization (DRO) method.
>
> In our related work, we described a technique that analyzes safety for DPO, and we remarked that it was far too restrictive in its goal (line 118). Any technique that is not studied to be generalizable to arbitrary objectives was too limited in its scope for us to compare to our method, which is fully generalizable to all auxiliary objectives. Our hypothesis is not only about safety - it is about **all possible reasonable auxiliary objectives**.
>
> Our conclusion on the safe RLHF paper is the same: it is neither demonstrably flexible nor computationally efficient and it hasn't been shown to generalize to other auxiliary objectives. Specifically, **safe RLHF simply uses PPO** (their paper: Appendix B.2) and comes with all of the efficiency issues that we mention on line 96. Based on Ethayarajh et al. (2024), we believe that aoPPO is a reasonable offline proxy for this technique, for which we provide results in Table 1/2.
>
> We have compared to three other multi-objective approaches that are well known and four preference-only baselines. While two of the multi-objective approaches are from other well-regarded publications, we augmented the best and most challenging offline RL-based preference-only baseline (oPPO) as our final multi-objective approach. We consistently showed that we outperformed all such benchmarks.
>
> **DRO is very similar to A-LOL from the perspective of policy optimization**. Given the advantage (rewards - values), the policy gradient is nearly the same (Alg 1 in DRO vs. Eqn 6 in A-LOL), except it seems that the regularization penalty for DRO is just the square of the penalty in A-LOL. We would expect DRO to perform similarly to A-LOL given that they are similar, and it is worth noting that A-LOL performs poorly (one of the most toxic).
> >Another concern is outdated models. I suggest using the new versions of the LLaMA, Mistral, or Gemma-2 models.
>
> Thank you for the suggestion. All of the models we leveraged are models from last year, and they are used in several works this year (for instance, Ethyarajh et al., 2024). There are several reasons why we chose these older models.
>
> - Newer techniques have begun to incorporate more and more safety measures (and other objectives) into their datasets and pre-training, which defeats the point of evaluating our safety alignment via an augmented objective. They are already significantly less toxic than models from last year, "pre-optimizing" for one of our auxiliary objectives and hence limiting the opportunity for the alignment methods we evaluate to make any measurable difference. For instance, Google pre-trains Gemma to "incorporat[e] comprehensive safety measures" already, and LLAMA-3 is significantly "more safe" than LLAMA due to "input safeguards". Since we want to most effectively evaluate alignment and **not** the pre-training of the LLM, we believe the older models provide a more rigorous challenge. If we were always able to pre-train or pre-tune for our alignment objectives, then there would be no necessity for alignment at all.
> - It allows us to easily detect scaling patterns on similar architectures (from 1.4B to 13B) because LLAMA and PYTHIA have widely-used variants across the model size spectrum (from small models to large models). To our knowledge, MISTRAL only has a 7B version. Gemma and newer LLAMA have many variants, but they are pre-tuned to be significantly more safe.
>
> **If it would significantly change your view on the paper, we could certainly demonstrate results on Gemma or others.** However, given the clear consensus across 5 model sizes and 2 model types, which is more than other multi-objective papers have shown, we do not believe a different result would be expected.
> > Lack of exploration on hyperparameters. DPO, KTO, and other optimization methods are very sensitive to different hyperparameters like beta, batch size, and learning rate.
>
> For all of our preference-only baselines, we mention that we use the exact hyperparameters used in the KTO paper (which they have tuned), which follow hyperparameters in the DPO paper (line 940, appendix). These have already been tuned with their best hyperparameters by the respective authors on the exact dataset that we use (which is the same combination as used in the KTO paper). **Hence, DPO and KTO and all other preference-only models are already fully hyperparameter tuned on the exact dataset we use.**
>
> For all the multiobjective baselines, we have specifically tuned their application-specific hyperparameters ourselves. MODPO is extremely sensitive to hyperparameters and required a small auxiliary weight ($w_0 = 1, w_1 < 0.5$), but for the others, we discovered that the hyperparameters from the original implementations worked best.

---

> > ### Comment · Reviewer_aNak · 2024-11-24
> >
> > Thanks for your explanation. I think the authors need to compare their method with safe-RLHF and update the results with new models. Note that current papers like SimPO show that preference optimization methods are sensitive to different hyper-parameters. I suggest the authors explore the performance of different methods on hyperparameters and report the best score for each method. I prefer to keep my score.

---

### Official Review · Reviewer_ya77 · 2024-11-04

**Soundness:** 3
**Presentation:** 4
**Contribution:** 3
**Rating:** 6
**Confidence:** 1

**Summary:**

The paper introduces a method called Hybrid Preference Optimization (HPO) to align large language models more effectively. HPO combines the efficiency of direct preference optimization (DPO) with the flexibility of reinforcement learning from human feedback (RLHF), enabling stable, computationally efficient training that focus on the capability of maximizing arbitrary non-differentiable and non-binary objectives.
The experimental results show that HPO outperforms traditional alignment methods, including DPO, RLHF, and other multi-objective approaches, in aligning language models with user preferences. HPO demonstrated marked improvements in optimizing auxiliary objectives, particularly for safety and readability, with lower violation rates on safety benchmarks and better readability scores compared to baselines.

**Strengths:**

The method is straightforward, requiring only a minor adjustment to KTO, yet it greatly enhances the optimization of key auxiliary objectives.

**Weaknesses:**

- The experiments are limited, focusing only on two objectives: reading level and sparse safety.
- Adding more objectives would exponentially increase the complexity of tuning the weights in Formula 15. Which is not effective.

**Questions:**

n/a

---

### Official Review · Reviewer_9ydw · 2024-11-06

**Soundness:** 2
**Presentation:** 2
**Contribution:** 2
**Rating:** 3
**Confidence:** 3

**Summary:**

While Direct Preference Optimization (DPO) is simpler and more stable than Reinforcement Learning from Human Feedback (RLHF), it falls short when it comes to incorporating arbitrary non-differentiable objectives. RLHF, particularly with on-policy algorithms like Proximal Policy Optimization (PPO), can be unstable and requires sampling from the language model during training, which is computationally expensive. The authors introduce Hybrid Preference Optimization (HPO) which addressees these issues by combining DPO and RLHF. HPO combines the simplicity of DPO with the flexibility of RLHF, allowing LLMs to be tuned using arbitrary auxiliary objectives without the need for on-policy generation. This hybrid approach leverages the strengths of both methods, aiming to improve the alignment of LLMs with both user preferences and designer-specified objectives.

**Strengths:**

1. The paper presents a novel method for integrating arbitrary auxiliary objectives into the DPO framework. This enhances the versatility of DPO, making it more practical for tuning LLMs to meet specific goals beyond user preferences.
2. In Section 4.1, the authors provide solid motivation for incorporating auxiliary rewards, backed by proofs and examples.
3. Implementing HPO requires only about 10 additional lines of code on top of the existing $\Psi$PO algorithm.

**Weaknesses:**

1. The paper frequently references $\Psi$PO and KTO but doesn't adequately explain these concepts in the preliminary sections. The writing is a bit hard to follow.
2. The method involves training an extra value network which adds to the computational load
3. HPO depends on manually defining and constructing auxiliary rewards. This process can be time-consuming and may require domain expertise.
4. Tables 2a, 2c, and 2d are not referred and properly discussed in the text.
5. The performance evaluation relies solely on assessments from GPT-4. Incorporating additional metrics, such as evaluations using reward models like ArmoRM, would provide a more comprehensive evaluation.
6. The paper doesn't include a Pareto analysis of different auxiliary rewards. This would provide understanding how the method balances multiple objectives and where trade-offs might occur.

**Questions:**

1. Could you explain what $L_2^{\tau}$ represents in Equation 12?
2. In Figure 4, what does "evaluation generation length relative to the chosen response" mean? Could you elaborate on this to clarify how it relates to your findings?

---

> ### Author Response · Authors · 2024-11-12
> **Rebuttal**
>
> Thank you for the feedback. Below are our responses to your comments.
>
> >The paper frequently references $\Psi$PO and KTO but doesn't adequately explain these concepts in the preliminary sections. The writing is a bit hard to follow.
>
> We apologise that the writing is hard to follow. However, to understand the inner-workings of KTO or $\Psi$PO is not fully necessary to comprehend our procedure (as it is in the main text): we believe the necessary information is provided on line 54 and in the associated references. We can essentially use any such preference learning technique as a blackbox (line 236) and simply apply an RL objective on the remaining auxiliary rewards. We've provided a comprehensive mathematical justification in Appendix B for why we can do that. For HPO, we do not assume any specifics of KTO beyond that it optimizes a preference objective (or a $\Psi$PO objective, which we use interchangeably), as shown in Appendix B.
>
> We are happy to improve the writing flow given specific feedback about what parts of the derivation and explanation of HPO are unclear.
>
> >The method involves training an extra value network which adds to the computational load
>
> We explicitly show in Table 5 and provide justification and analysis near line 490 about the computational load. Compared to the computational cost of the LLM, the value network is simply unnoticeable and insignificant. **Per example (averaged), it takes less than 0.5% of the time compared to the **cheapest** LLM (PYTHIA-1.4B) and it's less than 0.1% of the time compared to LLAMA-13B.**
>
> The additions in HPO are computationally insignificant compared to dataloading, forward passes through an LLM, or even worse, sampling from an LLM (which many RL techniques do, e.g., line 66).
>
> >HPO depends on manually [...] constructing auxiliary rewards. This process can be time-consuming and may require domain expertise.
>
> This is true of all optimization (you must define something to optimize), even outside of LLMs. The very existence of multi-objective optimization is predicated upon the premise that there exist multiple objectives to be optimized, and while it may be challenging to define them, it is still strictly superior to be able to optimize them than to not be able to do so.
>
> >Tables 2a, 2c, and 2d are not referred and properly discussed
>
> We reference the results in Table 2a in line 430 and 450, 2b on line 452, and 2c on lines 431 and 450. We do not explicitly mention all tables, but we discuss the overall auxiliary results (what is in Table 2) in detail with improvement margins between line 429 and line 453.
>
> Based on your feedback, we have added significantly more detail to these results for each of the subtables (in blue).
>
> >The performance evaluation relies solely on assessments from GPT-4. Incorporating additional metrics, such as evaluations using reward models like ArmoRM, would provide a more comprehensive evaluation.
>
> The evaluation relies on the **same reward models used for training** as well, which is an **exact proxy for how well the models were able to optimize the auxiliary objectives** (line 363). We show this evaluation in Table 2, where the reward models themselves indicate that HPO performs better than all other methods. GPT-4 is only a good proxy/judge for overall quality.
>
> Other proxies, such as ArmoRM, are inexact and do not necessarily correlate well with the auxiliary objective. The two questions we want to answer are (line 358, 363):
> - Can we actually optimize auxiliary objectives? For this, we use the same reward models used during training to evaluate whether the objective has truly been optimized (line 363). Even on this, the vast majority of multi-objective techniques CANNOT optimize our auxiliary objectives well (Table 2).
> - Can this optimization of auxiliary + preference objectives still yield good quality responses? This is where we use GPT-4.
> >Could you explain what L_2 represents in Equation 12?
>
> This is expectile regression with expectile $\tau$, with the same notation and usage as Kostrikov et al. (2021), provided as a reference on line 254/255.
>
> >In Figure 4, what does "evaluation generation length relative to the chosen response" mean? Could you elaborate on this to clarify how it relates to your findings?
>
> The evaluation dataset contains triplets of (prompt, chosen, rejected). We plot the length of the model's generation and divide it by the length of the chosen response ("gold label"). DPO and KTO essentially tend to ramble significantly (5-10x more) compared to the chosen response.
> >The paper doesn't include a Pareto analysis of different auxiliary rewards
>
> We don't have an explicit Pareto front visualization, but we ablate different weights for the two rewards in Table 9. Given those results, it is clear that our method **dominates** MODPO and A-LOL across the board (regardless of weight). **If you feel that it would significantly strengthen the paper, we can construct an explicit Pareto front showing HPO and aoPPO.**

---

> ### Comment · Reviewer_9ydw · 2024-11-25
> **Response to the authors**
>
> Thank you for addressing my questions. However, it appears that only minor revisions have been made to the manuscript, and I believe the writing could benefit from further improvement. Additionally, I don't see Table 9. I will maintain my original score.

---

### Note · Authors · 2024-11-25

I have read and agree with the venue's withdrawal policy on behalf of myself and my co-authors.